# Technical Note: Regression Analysis of Proximal Hyperspectral Data to Predict Soil pH and Olsen P

**Miles Grafton [1,](image)**, **Therese Kaul [1], Alan Palmer [1], Peter Bishop [1] and Michael White [2]**

[1]  School of Agriculture and Environment, Massey University, Private Bag, Palmerston North 4442, New Zealand; tmckaul@outlook.com (T.K.); a.s.palmer@massey.ac.nz (A.P.); p.a.bishop@massey.ac.nz (P.B.)

[2]  Ravensdown Fertiliser Ltd., P.O. Box 1049, Christchurch 8042, New Zealand; michael.white@ravensdown.co.nz

*  Correspondence: m.grafton@massey.ac.nz; Tel.: +64-6-356-9099

**Abstract:** This work examines two large data sets to demonstrate that hyperspectral proximal devices may be able to measure soil nutrient. One data set has 3189 soil samples from four hill country pastoral farms and the second data set has 883 soil samples taken from a stratified nested grid survey. These were regressed with spectra from a proximal hyperspectral device measured on the same samples. This aim was to obtain wavelengths, which may be proxy indicators for measurements of soil nutrients. Olsen P and pH were regressed with 2150 wave bands between 350 nm and 2500 nm to find wavebands, which were significant indicators. The 100 most significant wavebands for each proxy were used to regress both data sets. The regression equations from the smaller data set were used to predict the values of pH and Olsen P to validate the larger data set. The predictions from the equations from the smaller data set were as good as the regression analyses from the large data set when applied to it. This may mean that, in the future, hyperspectral analysis may be a proxy to soil chemical analysis; or increase the intensity of soil testing by finding markers of fertility cheaply in the field.

**Keywords:** regression analysis; soil fertility; correlation; statistical analysis; soil testing

---

## 1. Introduction

Fertilizer is the largest farm working expense in New Zealand hill country farming systems. Fertilizer applications are usually based on soil test results. This work seeks to establish if it is possible to measure soil fertility parameters using hyperspectral analysis. Large data sets are used to find the wave bands that may be proxies for laboratory testing [1].

New Zealand's hill country farming systems are largely based on sheep, beef, and sometimes deer animal production grazing on ryegrass (*Lolium perenne* L.) and white clover (*Trifolium repens* L.) swards. Easier topography exhibits higher concentrations of both, while harder hill country swards are dominated by species with lower feed value such as brown top (*Agrostis capillaries* L.) and crested dogstail (*Cynosurus cristatus* L.) with little clover. Steeper hill slopes tend to have higher concentrations of weed species such as (*Cirsium arvense* (L.) Scop.) and (*Ulex europaeus* L.).

New Zealand's temperate climate allows year round outside grazing. In the most severe environments, such as that created by altitude, animals are typically grazed on lower and better soils during winter and early spring. This is especially the case through lambing and calving in the South Island, and to a lesser extent the North Island's central plateau.

Sheep follow cattle in a grazing rotation as sheep graze shorter pasture than cattle and each animal class has different classes of parasites and parasitism is reduced as the parasite larvae of each class is destroyed by the grazing of the other. This farming system is heavily dependent on summer rainfall

that is quite variable, with higher rainfall on the West coast than the East coast of both islands [2,3], see
Figure 1. Where summer rainfall allows grass growth to finish stock farm incomes are higher than on
steeper, colder country, where in dry summers farmers are, forced to sell stock, unfinished as store, for
others to finish on softer country with irrigation [2]. On the steeper country, the use of vehicular traffic
is not possible, and fertilizer and herbicide applications are undertaken by aircraft [4].

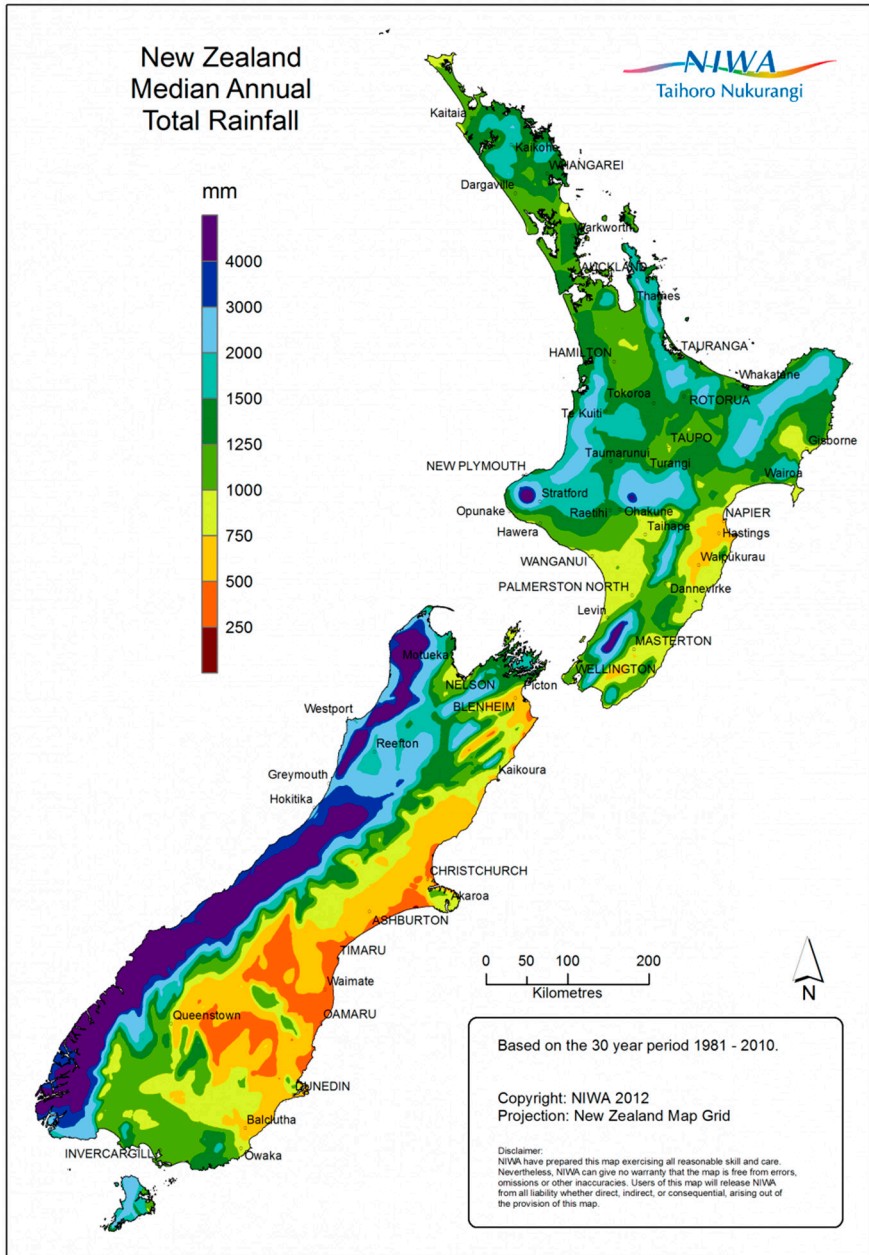

**Figure 1.** Shows, the location and 30-year, average annual rainfall for NZ (1981–2010). Source: National
Institute of Water and Atmosphere NZ. https://www.niwa.co.nz/climate/national-and-regional-
climate-maps/national accessed 7 March 2019.

Soil sampling for chemical analysis is expensive and limits the number of samples that it is
economic to analyze. This results in the averaging of soil samples over large areas up to several
hundred hectares for economies of scale [5]. The provision of an extremely large data set of soil
samples (the current price for a soil test is around US$70) made it possible to undertake some analysis
as this provided the actual data for this work. The cost of undertaking soil testing limits the intensity

and is another reason that samples are combined to provide an average. In addition, the samples were measured for spectral reflectance, which provided an opportunity to marry 2150 variables being the spectra reflectance values. This work was undertaken for another project so provided a unique opportunity to initiate work to see if soil elements can be predicted. In addition, a second data set was available at Massey, which had also been collected for other work, was available for testing [6]. The farms that were sampled would not have contemplated such intense soil testing if they were required to pay for it.

At present, on hill country farms, sampling is undertaken in transects that combine samples taken 10 m apart to average values from 27 samples [5]. Combining the samples eliminates the variation, especially for Olsen P. This test is the most important, in hill country fertilizer recommendations. The variation within transects is much greater than the 10 m sampling distance, therefore, there is no geospatial interpolation that is meaningful between sampling points [5–7]. Hyperspectral analysis has the potential to be a useful tool for proxy measurements of soil nutrients that could increase sampling density and reduce cost.

Hyperspectral analysis of soil using proximal visual and near infrared (VIS-NIR) spectra have been undertaken for some years. A review of the work indicates that it has been more useful in providing a proxy for soil organic matter and clay content than in predicting elements. However, clay compounds such as kaolinite and minerals such as calcite (a form of calcium carbonate) and pH, have been measured with some degree of success [8]. The proximal hyperspectral sensor used in this work is the Fieldspec 4 ASD (Analytical Spectral Devices, Malvern Panalytical, Malvern, UK), which has also had some success in mining large data sets for soil sensing [9–11]. These have shown that in clay, the total organic carbon and moisture have been measured, with variable accuracy, more so in soils with high soil organic matter [11]. Measurement of pH [9] and soil nitrogen [12] has been possible, but has been less robust in predictions.

This work aims to find the spectral wavebands that are proxies for Olsen P and pH, which are the two most important soil analyses used in New Zealand hill country farming. This is because plant available P reduces as soil acidity increases, acidic soil conditions are the norm in New Zealand. Measuring Olsen P using proximal hyperspectral analysis has not been successfully undertaken and pH with limited success.

## 2. Materials and Methods

### 2.1. Chemical Analyses

A data set of 3189 soil samples that had been analyzed, for full soil tests at Analytical Research Laboratory an ISO 17025 soil laboratory was married to spectra from the same soil samples collected using a probe on a Fieldspec 4. These measurements were undertaken at the Analytical Research Laboratory in Napier, New Zealand. The samples were from the four farms Ravensdown collected samples shown in Figure 2. These farms are part of a Private/Government research program between Ravensdown Ltd. and the New Zealand Ministry of Primary Industries, which requires extensive soil testing to 7.5 cm [13]. The project is to improve the performance of steep pastoral country and improve the efficiency of fertilizer delivery by aircraft. There is very little cropping and arable production on this land and all the samples are from pasture, which is the focus of the research. This was the original large data set used for the multiple partial least square regression (PLSR) using "R" 3.41 to identify wavebands of interest. Only Olsen P and pH were regressed as the second data set obtained independently had only Olsen P and pH measured at Massey University soil laboratory, which analyzed the 883 soil samples. The smaller Massey data set was sampled at depths (0–3 cm, 3–15 cm, and 15–30 cm). Whereas, all samples from the larger Ravensdown data set were at (0–7.5 cm).

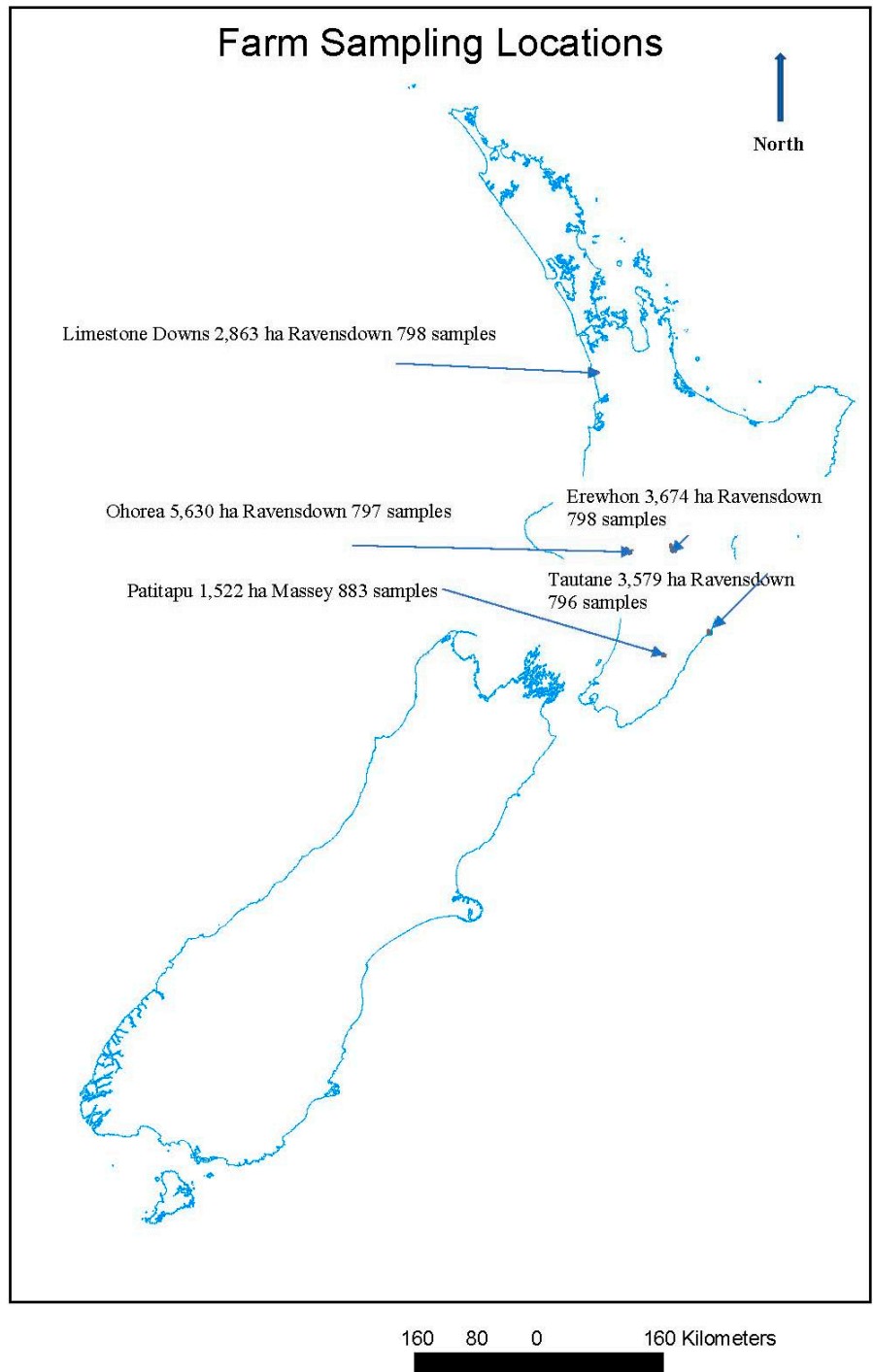

**Figure 2.** Location and name of farms where samples were collected. The sampling body (Ravensdown Ltd, or Massey University) the size of the farm and number of samples taken.

The second data set was from two stratified nested grid samples from different fields on the same farm used to look at geospatial variance with soil depth [6,7]. The samples were tested for Olsen P using the chemical extraction methods and photo-spectrometer was used for the geospatial analysis [6,7]. Those that had not already been measured were completed using the method employed in References [7,14]. The pH measurements were taken using a digital probe in 10 g samples in 25 mL of distilled water. All Fieldspec 4 ASD measurements, at Massey and in Napier were on dry samples to eliminate the effects of moisture [15].

## 2.2. Statististical Analysis

A PLSR of 2150 wavebands of spectra data from the Fieldspec 4 (350–2500 nm) of 3189 samples were regressed against the chemical analysis results. A PLSR assumes the data to be normally distributed, the mean and standard deviations of the data sets are presented in Table 1. The shape of the distributions are shown in Figure 3a–d.

**Table 1.** Mean and standard deviations of the two data sets.

| Data Set | Mean | Standard Deviation |
|---|---|---|
| Massey Olsen P | 18.1 | 13.9 |
| Massey pH | 5.55 | 0.33 |
| Ravensdown Olsen P | 23.1 | 15.4 |
| Ravensdown pH | 5.72 | 0.32 |

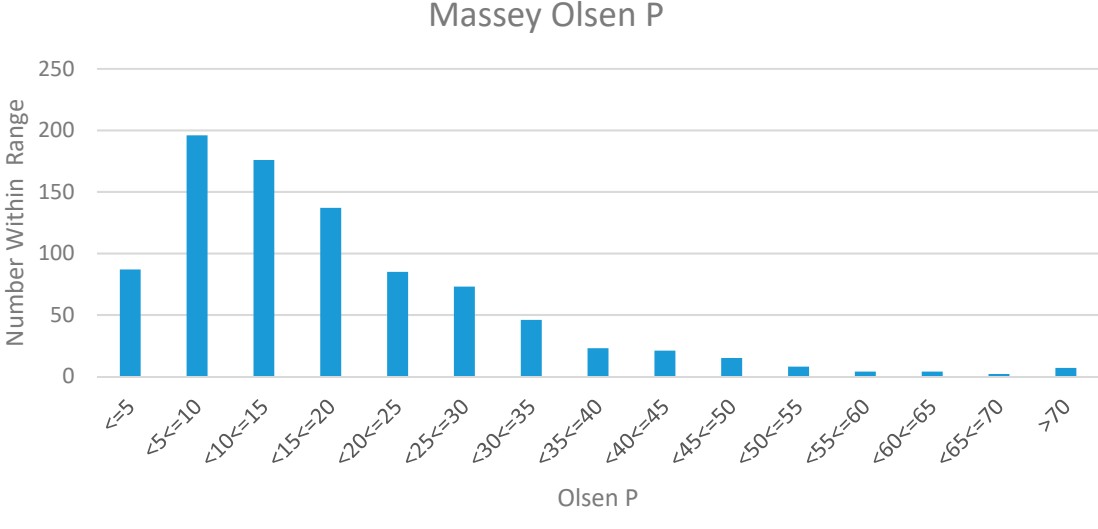

(a)

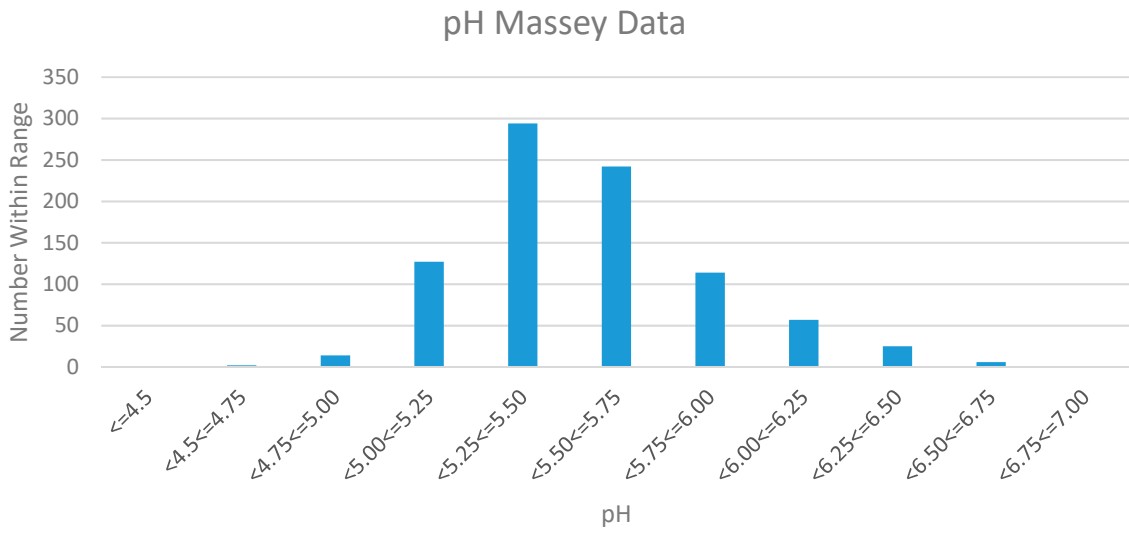

(b)

**Figure 3.** *Cont.*

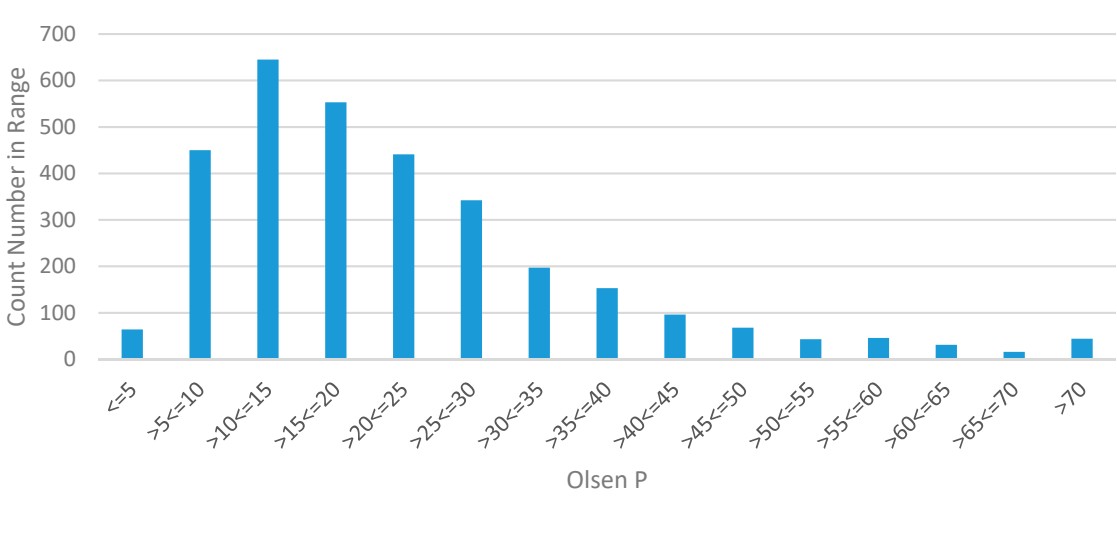

(**c**)

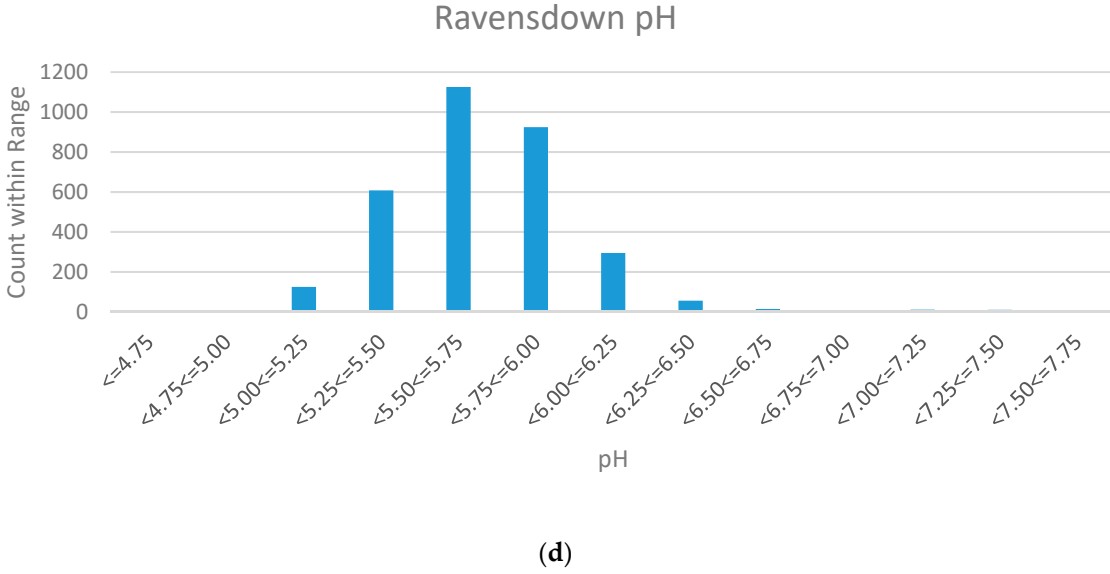

(**d**)

**Figure 3.** (**a**) Shows the distribution of the Massey Olsen P data; (**b**) shows the distribution of Massey pH data; (**c**) shows the distribution of Ravensdown Olsen P data; and (**d**) shows the distribution of Ravensdown pH data.

The Olsen P distributions for both data sets resembled a Chi$^2$ more than a normal distribution. For this reason a log$_{10}$ of Olsen P of the Ravensdown data and Massey data was regressed as this passed the normality test (see Figure 4).

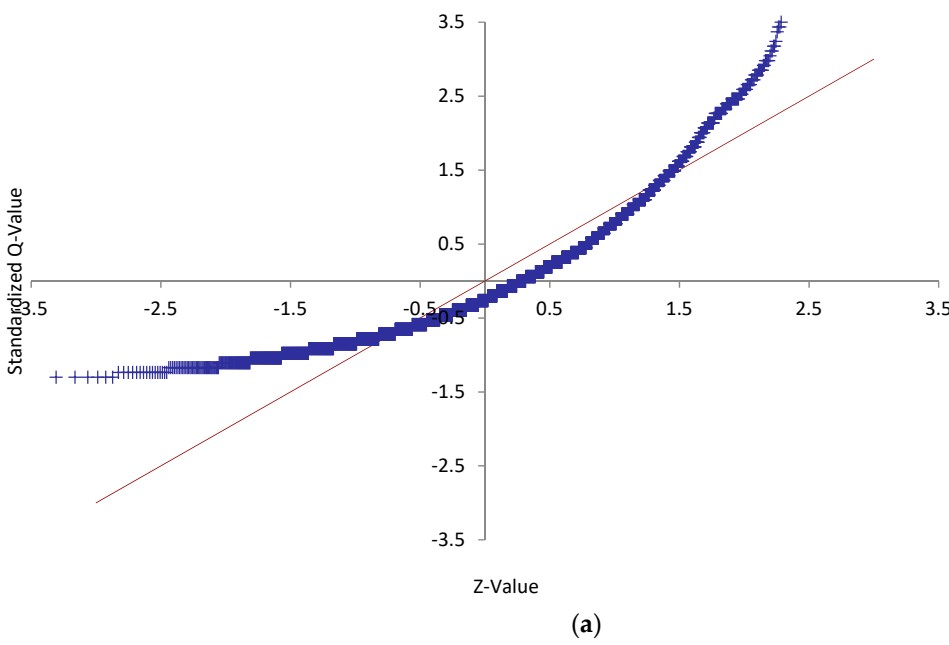

(**a**)

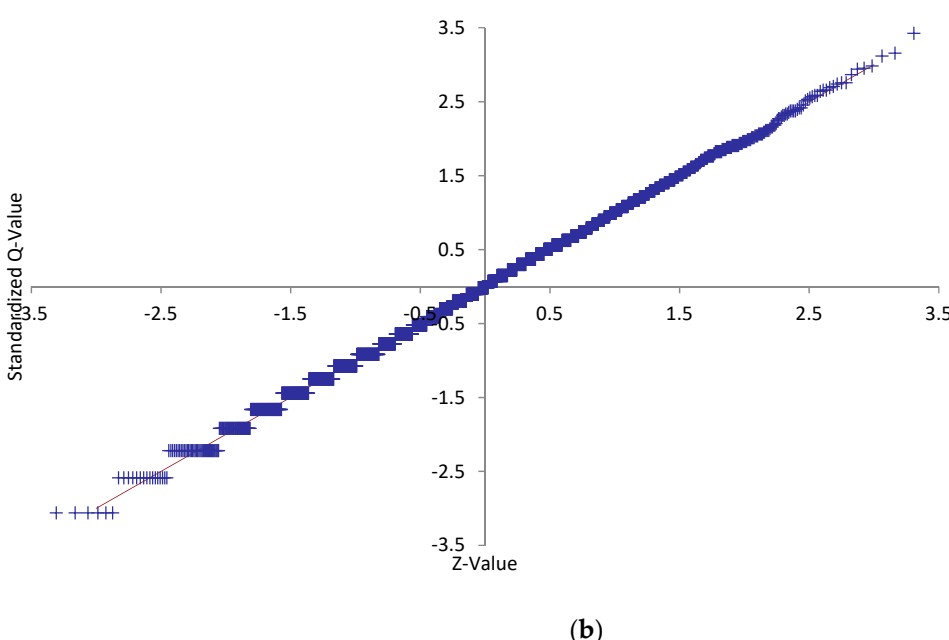

(**b**)

**Figure 4.** (**a**) shows a Q-Q Normality test for the raw Olsen P data with a Chi$^2$ distribution. (**b**) Shows that the Log$_{10}$ of the Olsen P of the data produces a normal distribution.

For this work, the larger data set was regressed using "R" version 3.41 [11,16,17]. The 100 most significant wavebands were selected for Olsen P and pH to regress the smaller Massey University data set. This is because the Massey data set had 883 samples so the number of waveband (factors) to be regressed against as the independent variables should be a much smaller quantity. The regression was undertaken initially using "R", then repeated using an Excel plug in StatTools version 7.5 (Pallisade, NY, USA). This was undertaken to check that the results were the same.

The regression equations from the Massey University data set were then used to predict Olsen P and pH on the large dataset. The predictions were then compared to the actual laboratory values. This was to establish if the regression of a large data set had found a proxy means of measuring pH and Olsen P using independent data sets.

## 3. Results

The regression analyses on the data sets for pH and Olsen P, summarized in Table 2, for Ravensdown Ltd. and Massey University data, showed a significant trend between the modeled and actual values for all data set ($p$ value < 0.0001). However, there is a high degree of variability within the data sets shown by the high standard errors and low adjusted $R^2$ values, with the best predictions being for pH. The summary of the scatter plots obtained from the regression are in Figure 5a–f.

**Table 2.** Summary of regression analyses.

| Data Set | Adj. $R^2$ | Std. Err | F | $p$ Value |
|---|---|---|---|---|
| Ravensdown Olsen P [1] | 0.47 | 11.21 | 2.319 | <2.2 × 10$^{-16}$ |
| Ravensdown pH [1] | 0.71 | 0.17 | 4.688 | <2.2 × 10$^{-16}$ |
| Massey pH [2] | 0.42 | 0.25 | 7.3 | <2.2 × 10$^{-16}$ |
| Massey pH [3] | 0.42 | 0.25 | 7.3 | <0.0001 |
| Massey Olsen P [3] | 0.56 | 9.03 | 11.99 | <0.0001 |
| Ravensdown pH [4] | 0.42 | 0.24 | 24.53 | <0.0001 |
| Ravensdown Olsen P [4] | 0.27 | 13.21 | 12.53 | <0.0001 |
| Ravensdown Log OP [4] | 0.34 | 9.03 | 17.42 | <0.0001 |
| Massey Log Olsen P [4] | 0.46 | 0.25 | 8.5 | <0.0001 |

[1] R regression 2148 and 1040 degrees of freedom; [2] R regression 100 and 782 degrees of freedom; [3] StatTools regression 100 and 782 degrees of freedom; and [4] StatTools regression 100 and 3088 degrees of freedom.

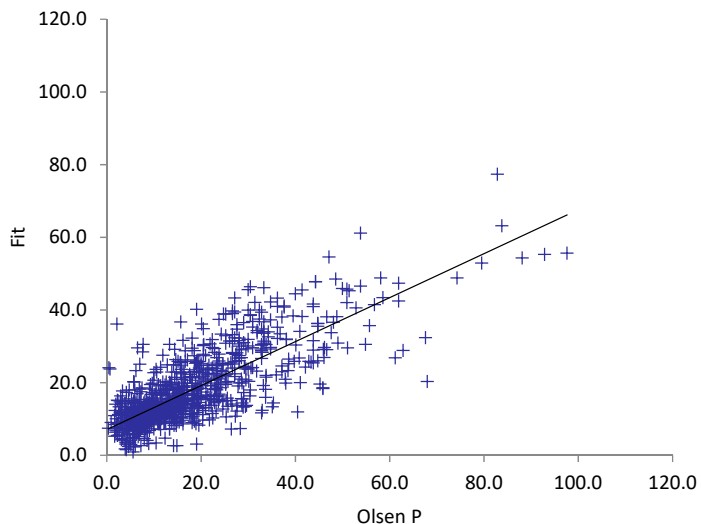

(**a**) Regression of Massey Olsen P data

**Figure 5.** *Cont.*

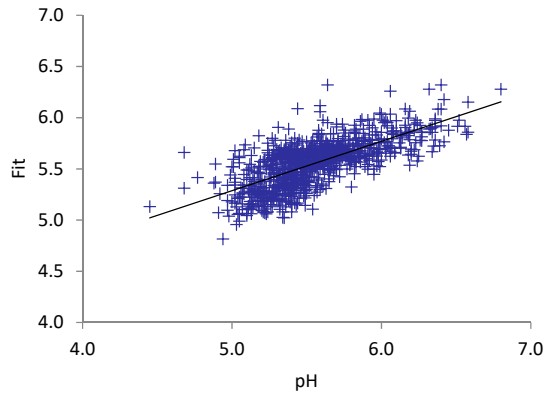

(**b**) Regression of Massey pH data

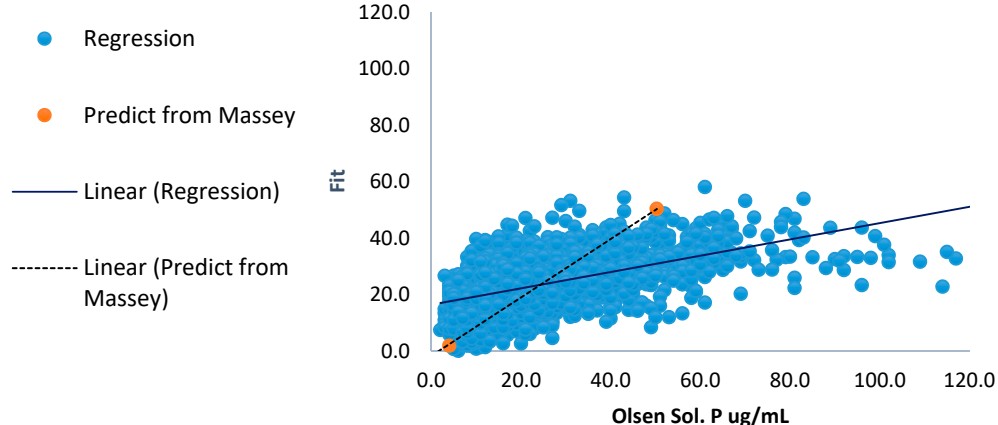

(**c**) Regression of Ravensdown Olsen P data and prediction using the equation from Massey "a"

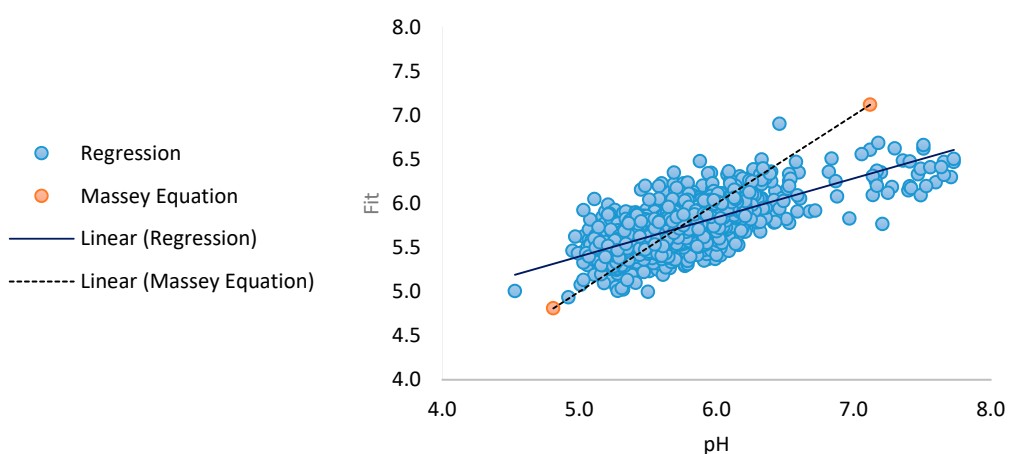

(**d**) Regression of Ravensdown pH data and prediction using the equation from Massey "b"

**Figure 5.** *Cont.*

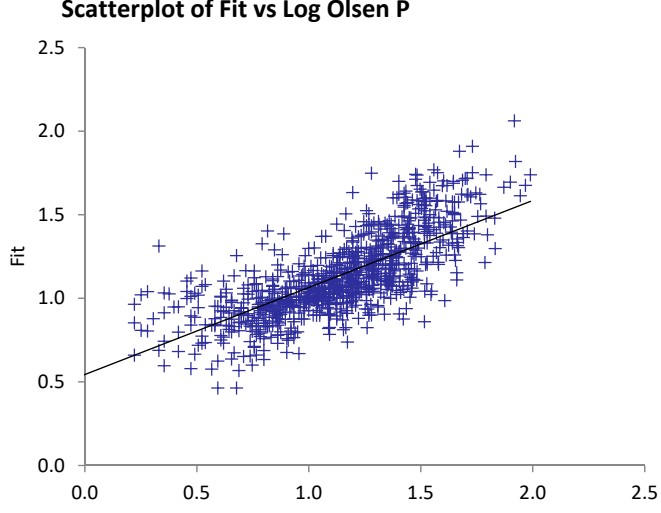

(**e**) Regressionof Log of Olsen P of Massey data

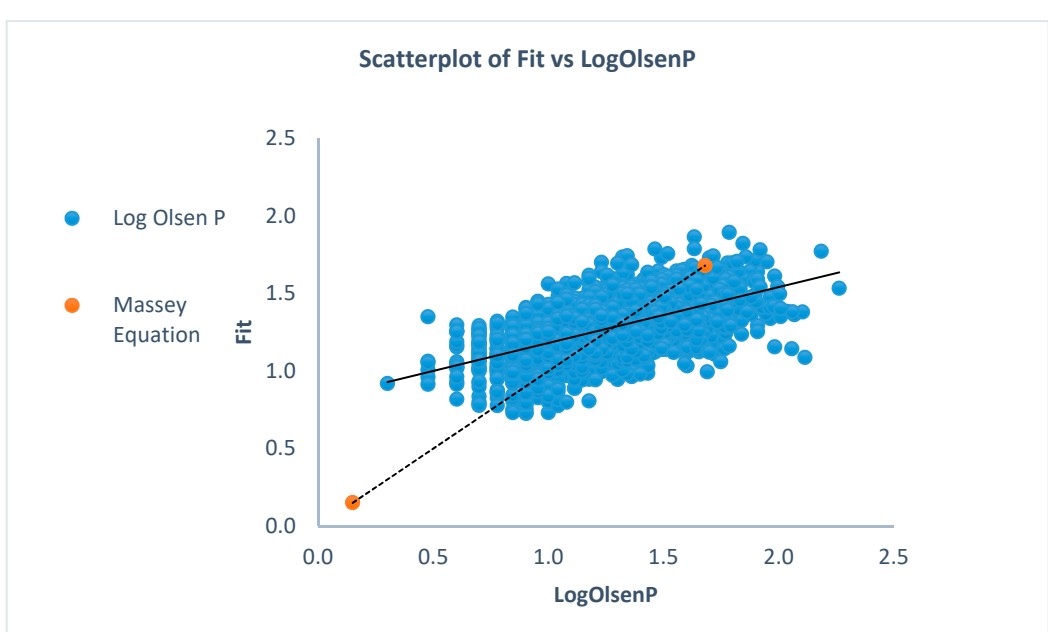

(**f**) Regression of Log of Olsen P of Ravensdown data and Ravensdown data using the Massey Log of Olsen P equation

**Figure 5.** Scatter plots of Olsen P and pH modelled results compared to measured, for (**a**,**b**) Massey calibration set and (**c**–**f**) application of the Massey model to the Ravensdown set of data for validation and comparison to the data set regression.

The Ravensdown regression analyses have predictions using the regression equations developed on the Massey data. The number of wavebands was reduced from 2150 to 100 to reduce auto correlation and co-linearity effects from the regression, which are inevitable from such a large data set; especially if moving to a smaller data set with less rows than columns used in the PLSR. The significance of waveband contribution were all significant less than 0.05. The regression analyses using R on the Ravensdown data suggest that even though there are more rows than columns regressing using 2150 columns has an auto correlation effect (see Table 2). The equations from the analyses appear in Appendix A.

## 4. Discussion

The Massey soil data came from 30 cm soil cores sectioned 0–3 cm, 3–15 cm, and 15–30 cm soil depths, which contained samples from the lowest strata with extremely small levels of Olsen P some samples having undetectable levels. This meant that the Massey data regression for Olsen P passes close to the origin. Olsen P is a measure of exchangeable P and is readily available for plant uptake, while the remainder of the P in the soil is in less available forms organic (biomass/animal excrement) and complexed with Al and Fe. The stratification of phosphate in the soil in closely related to the organic matter stratification. The most popular fertilizer applied in New Zealand is single super phosphate $(Ca(H_2PO_4)_2)$ and the next most popular fertilizer supplying P is diammonium phosphate $((NH_4)_2HPO_4)$ [18–20]. Phosphates are not very soluble, which can be problematic in situations where surface run-off is likely [20]. The majority of phosphate tends to remain near the surface of soil horizons through cation exchange with clay minerals, which is why soil samples are taken to 7.5 cm in New Zealand pastures [5,19]. While the hyperspectral data was regressed against Olsen P, the spectra reflectance will be on compounds containing phosphate, similarly to calcium being found in calcite [8].

The regression of the Massey Olsen P data had an adjusted $R^2$ of 0.55 whereas the Ravensdown data produced an adjusted $R^2$ of 0.27. Both data sets had high variation in data fit, but the trends from the analyses are highly significant. The Massey data probably produced a clearer trend as the stratified sample had a larger range of values with increasing values as samples neared the surface [6,7]. The Ravensdown regression improved to an $R^2$ of 0.34 when the $\log_{10}$ of Olsen P was regressed. These results from the Chi$^2$ shape of the raw data became more normalized when the log values were regressed.

The pH data was more normally distributed than the Olsen P data for both data sets. Both data sets were able to be regressed to the hyperspectral data with similar trends, which were highly significant. The $R^2$ of the Massey and Ravensdown data were about 0.42.

The regression equations found from the PLSR of the Massey data were also used to predict Olsen P and pH of the Ravensdown data, as shown in Figure 5c,d,f. This exercise was undertaken to see if the equations developed from one data set were able to provide a reasonable fit with the other data set when using the same wavebands. While the gradients of the equations in Figure 5c,d,f are quite different; they both bisect the data values almost in half. This suggests that the spectral wavebands selected may be transferable to other data sets and that further research and analysis using these wavebands may be worthwhile.

This exercise shows that there is promise in developing proximal hyperspectral sensing of soil nutrients. This may prove to be a cheaper alternative to chemical analysis if it can be undertaken in situ in the field. This could lead to more intensive testing which would be a benefit in delivering the desired fertilizer nutrient to where it is needed through variable rate application, providing more efficient fertilizer utilization.

This study does demonstrate that taking a large data set from a large range of soil types from many locations over thousands of hectares can produce promising results with statistical significance. Other work in this field has been on much smaller data sets applied in localized areas on similar soils. The results of this study are as good as other work and are a step into finding minerals or proxies for fertility.

**Author Contributions:** M.W. conceived of the research in discussion with M.G. T.K. undertook the chemical analysis at Massey University. P.B., M.G. and A.P. assisted with sample preparation. M.W. provided the Ravensdown Ltd. data set. T.K. undertook the hyperspectral analysis of the Massey data set. M.G. undertook the statistical analysis. M.G. edited the paper in discussion with the co-authors.

**Acknowledgments:** This study was funded by, Ravensdown Ltd. and Massey University. Ravensdown paid for the preparation and analysis of a data set, whilst Massey University paid for and undertook the analysis of a second independent data set. This research received no external funding.

**Conflicts of Interest:** The authors declare no conflict of interest.

## Appendix A

The regression equations where ($\times\times\times$ nm) stands for the reflectance values for the wavebands from the Fieldspec 4. The predicted Olsen P appears on the y axis using the StatTools software.

$$
\begin{aligned}
\text{Olsen P Massey Data} ={}& 17.39357943 + 924.99461634 \times 515 \text{ nm} - 1023.05662383 \times 516 \\
& \text{nm} + 839.64318782 \times 549 \text{ nm} - 1901.74299473 \times 550 \text{ nm} + 1137.32505077 \times 551 \text{ nm} - \\
& 1574.15784614 \times 611 \text{ nm} - 2656.44564275 \times 612 \text{ nm} - 1236.46471204 \times 613 \text{ nm} + \\
& 713.64885867 \times 614 \text{ nm} + 4529.23578307 \times 626 \text{ nm} - 939.78101407 \times 627 \text{ nm} - \\
& 629.35692109 \times 628 \text{ nm} + 3612.28340339 \times 629 \text{ nm} + 1915.06008321 \times 630 \text{ nm} + \\
& 255.93665506 \times 662 \text{ nm} - 7544.07713423 \times 663 \text{ nm} + 638.25334103 \times 664 \text{ nm} + \\
& 5506.80104833 \times 689 \text{ nm} - 1248.88851626 \times 690 \text{ nm} - 115.82768303 \times 708 \text{ nm} - \\
& 1517.88015709 \times 709 \text{ nm} - 2718.51566921 \times 718 \text{ nm} + 400.99891055 \times 727 \text{ nm} + \\
& 1629.99158725 \times 728 \text{ nm} + 1604.20664865 \times 729 \text{ nm} - 1048.07981365 \times 730 \text{ nm} + \\
& 396.39340421 \times 757 \text{ nm} - 433.694892 \times 758 \text{ nm} + 3035.53288044 \times 759 \text{ nm} - \\
& 614.25777917 \times 762 \text{ nm} - 1551.48090776 \times 802 \text{ nm} - 489.15224109 \times 803 \text{ nm} - \\
& 1372.19196525 \times 804 \text{ nm} - 2225.00027627 \times 866 \text{ nm} + 3936.97208447 \times 867 \text{ nm} + \\
& 61.51888687 \times 868 \text{ n} - 514.77643709 \times 1178 \text{ nm} - 1017.23089936 \times 1179 \text{ nm} + \\
& 1490.52821841 \times 1232 \text{ nm} + 422.56714205 \times 1483 \text{ nm} + 1043.34099139 \times 1484 \text{ nm} + \\
& 1378.72044837 \times 1485 \text{ nm} - 1762.26783483 \times 1486 \text{ nm} - 110.38296651 \times 1487 \text{ nm} - \\
& 1606.86292061 \times 1488 \text{ nm} + 420.02860207 \times 1489 \text{ nm} - 1060.13876294 \times 1490 \text{ nm} + \\
& 720.27928825 \times 1491 \text{ nm} - 28.76362342 \times 1492 \text{ nm} + 103.28184426 \times 1493 \text{ nm} - \\
& 131.95597966 \times 1494 \text{ nm} - 365.83963852 \times 1793 \text{ nm} - 553.20867642 \times 1886 \text{ nm} + \\
& 759.70141479 \times 1887 \text{ nm} + 1193.69613109 \times 1888 \text{ nm} - 3810.90239696 \times 1898 \text{ nm} + \\
& 1371.82998738 \times 1899 \text{ nm} + 2366.01957159 \times 1907 \text{ nm} - 1385.13255379 \times 1908 \text{ nm} + \\
& 1389.98364401 \times 1911 \text{ nm} + 1686.25002952 \times 1912 \text{ nm} - 590.34003403 \times 1914 \text{ nm} - \\
& 2568.58189163 \times 1916 \text{ nm} + 1891.41352763 \times 1925 \text{ nm} - 3160.17813263 \times 1940 \text{ nm} + \\
& 1545.65998348 \times 1941 \text{ nm} + 1350.36874433 \times 1945 \text{ nm} - 1110.56764511 \times 1956 \text{ nm} + \\
& 466.15005224 \times 1989 \text{ nm} - 263.86092155 \times 1993 \text{ nm} - 1999.25991196 \times 2068 \text{ nm} + \\
& 682.72535112 \times 2069 \text{ nm} - 1300.80069545 \times 2070 \text{ nm} + 2035.22693592 \times 2071 \text{ nm} - \\
& 533.64076109 \times 2083 \text{ nm} + 1647.95747007 \times 2084 \text{ nm} - 1711.27378902 \times 2192 \text{ nm} + \\
& 1915.97472243 \times 2193 \text{ nm} - 460.02914387 \times 2360 \text{ nm} - 1850.76561572 \times 2364 \text{ nm} - \\
& 2245.58661469 \times 2365 \text{ nm} + 3347.65479354 \times 2368 \text{ nm} + 648.05853418 \times 2382 \text{ nm} + \\
& 15.93562167 \times 2385 \text{ nm} + 34.51065143 \times 2386 \text{ nm} - 158.51284072 \times 2389 \text{ nm} + \\
& 1947.3063904 \times 2390 \text{ nm} - 1145.3684943 \times 2394 \text{ nm} - 3335.29857385 \times 2426 \text{ nm} + \\
& 3554.50781574 \times 2427 \text{ nm} - 2277.58894701 \times 2428 \text{ nm} + 3403.1296183 \times 2429 \text{ nm} + \\
& 1109.2586938 \times 2430 \text{ nm} - 3084.81572109 \times 2431 \text{ nm} + 1323.24564451 \times 2432 \text{ nm} + \\
& 3395.88193144 \times 2433 \text{ nm} - 5697.99831042 \times 2434 \text{ nm} + 3244.05960391 \times 2450 \text{ nm} - \\
& 695.62046814 \times 2469 \text{ nm} - 1721.5855356 \times 2470 \text{ nm}
\end{aligned}
$$

(A1)

$$
\begin{aligned}
\text{pH Massey data} = {} & 5.63010896 + 1.01794868 \times 359\text{ nm} + 11.21144696 \times 360\text{ nm} - \\
& 7.32275159 \times 361\text{ nm} - 21.40383301 \times 389\text{ nm} - 14.03322979 \times 390\text{ nm} + 30.74610746 \\
& \times 391\text{ nm} - 43.53523382 \times 392\text{ nm} + 18.23671329 \times 394\text{ nm} - 6.38500327 \times 395\text{ nm} + \\
& 18.01733161 \times 396\text{ nm} - 27.36353051 \times 432\text{ nm} - 19.20056161 \times 439\text{ nm} + 92.96690364 \\
& \times 461\text{ nm} + 48.16287377 \times 516\text{ nm} - 43.75929762 \times 517\text{ nm} - 45.84482625 \times 518\text{ nm} - \\
& 16.79685965 \times 522\text{ nm} - 6.78723094 \times 536\text{ nm} - 25.46011315 \times 537\text{ nm} + 68.40619006 \\
& \times 538\text{ nm} - 24.08077525 \times 576\text{ nm} - 16.97404877 \times 577\text{ nm} + 8.06429782 \times 578\text{ nm} - \\
& 67.30041419 \times 579\text{ nm} - 6.13309954 \times 580\text{ nm} + 38.00080165 \times 581\text{ nm} + 19.49066231 \\
& \times 596\text{ nm} + 57.02874678 \times 597\text{ nm} + 56.66581798 \times 598\text{ nm} + 40.13382241 \times 608\text{ nm} + \\
& 11.51929348 \times 609\text{ nm} - 5.23723548 \times 610\text{ nm} - 115.34856771 \times 611\text{ nm} + 27.83241776 \\
& \times 634\text{ nm} - 21.40988878 \times 685\text{ nm} - 66.58131083 \times 686\text{ nm} + 37.08261582 \times 701\text{ nm} + \\
& 20.93483536 \times 702\text{ nm} + 25.60900836 \times 703\text{ nm} - 16.63755709 \times 704\text{ nm} - 7.58450299 \\
& \times 731\text{ nm} - 73.19155201 \; 732\text{ nm} + 3.76616034 \times 733\text{ nm} + 9.37274134 \times 767\text{ nm} + \\
& 56.97927421 \times 768\text{ nm} + 43.02161764 \times 769\text{ nm} - 10.15430763 \times 784\text{ nm} - 34.43326738 \\
& \times 785\text{ nm} - 26.49887985 \times 786\text{ nm} + 129.32222188 \times 891\text{ nm} + 3.23990609 \times 929\text{ nm} - \\
& 68.25542179 \times 930\text{ nm} - 45.04527251 \times 931\text{ nm} + 13.84837643 \times 1100\text{ nm} - 7.99616613 \\
& \times 1101\text{ nm} + 51.69035257 \times 1109\text{ nm} - 7.9522317 \times 1110\text{ nm} - 58.35765755 \times 1111\text{ nm} \\
& + 64.85300645 \times 1115\text{ nm} + 30.3465026 \times 1149\text{ nm} - 105.98328731 \times 1170\text{ nm} - \\
& 31.56912981 \times 1248\text{ nm} + 53.76128383 \times 1251\text{ nm} + 21.86762012 \times 1255\text{ nm} - \\
& 41.91309234 \times 1259\text{ nm} - 22.43674708 \times 1264\text{ nm} + 57.78670883 \times 1265\text{ nm} - \\
& 67.27240656 \times 1273\text{ nm} + 34.20628164 \times 1274\text{ nm} - 45.64447098 \times 1276\text{ nm} + \\
& 9.64816955 \times 1277\text{ nm} - 3.06623343 \times 1278\text{ nm} + 29.37988671 \times 1280\text{ nm} + \\
& 36.03037732 \times 1281\text{ nm} - 71.25645933 \times 1284\text{ nm} + 6.79885619 \times 1285\text{ nm} + \\
& 29.11077935 \times 1288\text{ nm} + 42.77467661 \times 1293\text{ nm} - 23.89894933 \times 1294\text{ nm} - \\
& 1.03098712 \times 1297\text{ nm} - 38.81837219 \times 1300\text{ nm} + 61.98138819 \times 1301\text{ nm} + \\
& 42.9406813 \times 1304\text{ nm} + 37.08906097 \times 1305\text{ nm} - 18.55261338 \times 1308\text{ nm} - \\
& 82.81463312 \times 1312\text{ nm} - 11.28607026 \times 1418\text{ nm} + 45.80234711 \times 1421\text{ nm} - \\
& 17.00111204 \times 1422\text{ nm} - 81.73345042 \times 1424\text{ nm} + 36.98733349 \times 1425\text{ nm} + \\
& 92.21986869 \times 1426\text{ nm} - 57.31536538 \times 1434\text{ nm} + 47.75592428 \times 1641\text{ nm} - \\
& 110.00831886 \times 1645\text{ nm} + 69.69764291 \times 1649\text{ nm} - 4.20631941 \times 1889\text{ nm} - \\
& 23.3992297 \times 2209\text{ nm} + 58.41679711 \times 2218\text{ nm} - 34.627777 \times 2229\text{ nm}
\end{aligned}
\tag{A2}
$$

$$
\begin{aligned}
\text{Ravensdown Olsen Sol. P ug/mL} =\ & 30.23032375 - 17395.35198111 \times 515\ \text{nm} \\
& + 16969.34406014 \times 516\ \text{nm} + 3377.61981422 \times 549\ \text{nm} + 52052.72258803 \times 550\ \text{nm} \\
& - 56378.23039846 \times 551\ \text{nm} - 10883.92875306 \times 611\ \text{nm} - 2804.46156902 \times 612\ \text{nm} \\
& + 31769.66357798 \times 613\ \text{nm} - 17718.32239769 \times 614\ \text{nm} - 35540.91236295 \times 626\ \text{nm} \\
& + 19983.25638365 \times 627\ \text{nm} + 37138.05492859 \times 628\ \text{nm} - 42450.34402066 \times 629\ \text{nm} \\
& + 29197.99472892 \times 630\ \text{nm} - 46675.44488017 \times 662\ \text{nm} + 94780.04643109 \times 663\ \text{nm} \\
& - 64090.24301552 \times 664\ \text{nm} + 29987.66554762 \times 689\ \text{nm} - 16542.41570829 \times 690\ \text{nm} \\
& - 24524.92814083 \times 708\ \text{nm} + 16852.13726875 \times 709\ \text{nm} + 13915.91992942 \times 718\ \text{nm} \\
& - 261970.26369825 \times 727\ \text{nm} + 475840.57721582 \times 728\ \text{nm} - 391631.84031072 \times 729\ \text{nm} \\
& + 167683.42635072 \times 730\ \text{nm} + 34174.1301415 \times 757\ \text{nm} - 55664.05487287 \times 758\ \text{nm} \\
& + 45346.43952343 \times 759\ \text{nm} - 24529.90590256 \times 762\ \text{nm} - 56908.44102923 \times 802\ \text{nm} \\
& + 68173.50089467 \times 803\ \text{nm} - 10648.33817098 \times 804\ \text{nm} - 45812.18514267 \times 866\ \text{nm} \\
& + 44779.116385 \times 867\ \text{nm} + 148.20876873 \times 868\ \text{nm} + 21509.19608288 \times 1178\ \text{nm} \\
& - 23210.76230177 \times 1179\ \text{nm} + 1205.11444786 \times 1232\ \text{nm} - 21251.86693495 \times 1483\ \text{nm} \\
& + 124470.56089342 \times 1484\ \text{nm} - 105176.50856368 \times 1485\ \text{nm} - 185934.27474499 \times 1486 \\
& \text{nm} + 138981.68049312 \times 1487\ \text{nm} + 246634.81183744 \times 1488\ \text{nm} - 328405.42442083 \times \\
& 1489\ \text{nm} + 347007.55586028 \times 1490\ \text{nm} - 292201.5362165 \times 1491\ \text{nm} + 200837.6178813 \\
& \times 1492\ \text{nm} - 179815.78120518 \times 1493\ \text{nm} + 55726.26983619 \times 1494\ \text{nm} - 16.73830056 \\
& \times 1793\ \text{nm} - 191.46145396 \times 1886\ \text{nm} - 15495.92099449 \times 1887\ \text{nm} + 20352.7752412 \times \\
& 1888\ \text{m} - 19778.9353901 \times 1898\ \text{nm} + 13569.86772063 \times 1899\ \text{nm} + 13684.31307497 \times \\
& 1907\ \text{nm} - 4213.77774362 \times 1908\ \text{nm} - 50337.04369486 \times 1911\ \text{nm} + 47183.0215495 \times \\
& 1912\ \text{nm} - 7816.0179793 \times 1914\ \text{nm} + 5392.85376083 \times 1916\ \text{nm} - 2052.78826421 \times \\
& 1925\ \text{nm} + 17278.11062412 \times 1940\ \text{nm} - 27761.29324011 \times 1941\ \text{nm} + 5853.66779632 \times \\
& 1945\ \text{nm} + 5254.85824641 \times 1956\ \text{nm} + 3364.2802088 \times 1989\ \text{nm} - 3917.88758909 \times \\
& 1993\ \text{nm} - 56380.67020239 \times 2068\ \text{nm} + 60708.68293837 \times 2069\ \text{nm} - 7333.4871515 \times \\
& 2070\ \text{nm} - 1562.02303576 \times 2071\ \text{nm} + 15196.53356113 \times 2083\ \text{nm} - 11715.24142948 \times \\
& 2084\ \text{nm} + 713.06495212 \times 2192\ \text{nm} - 1071.47197067 \times 2193\ \text{nm} - 2614.91400997 \times \\
& 2360\ \text{nm} + 11410.26614702 \times 2364\ \text{nm} - 7602.59475447 \times 2365\ \text{nm} + 329.78450793 \times \\
& 2368\ \text{nm} - 697.88676336 \times 2382\ \text{nm} - 5002.8475417 \times 2385\ \text{nm} + 6260.27394336 \times \\
& 2386\ \text{nm} - 6485.55173936 \times 2389\ \text{nm} + 3766.47869786 \times 2390\ \text{nm} - 534.76176827 \times \\
& 2394\ \text{nm} - 651.09384695 \times 2426\ \text{nm} + 14459.57991864 \times 2427\ \text{nm} - 8263.76328712 \times \\
& 2428\ \text{nm} - 18615.1875937 \times 2429\ \text{nm} + 20821.74215637 \times 2430\ \text{nm} - 22094.66916225 \times \\
& 2431\ \text{nm} + 19958.35600318 \times 2432\ \text{nm} + 5688.84328148 \times 2433\ \text{nm} - 9439.06667547 \times \\
& 2434\ \text{nm} - 337.40531342 \times 2450\ \text{nm} + 1567.16270499 \times 2469\ \text{nm} - 1289.79561673 \times \\
& 2470\ \text{nm}
\end{aligned}
$$

(A3)

$$
\begin{aligned}
\text{Ravensdown pH} =\ & 6.18674446 - 1.69869607 \times 359\ \text{nm} + 6.31484627 \times 360\ \text{nm} + \\
& 0.08986659 \times 361\ \text{nm} - 10.19718185 \times 389\ \text{nm} + 57.97841255 \times 390\ \text{nm} - 88.63455731 \\
& \times 391\ \text{nm} + 29.6510304 \times 392\ \text{nm} - 3.20305907 \times 394\ \text{nm} + 9.54930601 \times 395\ \text{nm} - \\
& 1.69283658 \times 396\ \text{nm} + 20.93524516 \times 432\ \text{nm} - 6.77887439 \times 439\ \text{nm} - 45.13080489 \times \\
& 461\ \text{nm} - 173.58845438 \times 516\ \text{nm} + 266.99997247 \times 517\ \text{nm} - 385.88722915 \times 518\ \text{nm} \\
& + 287.1056356 \times 522\ \text{nm} - 254.31109064 \times 536\ \text{nm} + 647.91459662 \times 537\ \text{nm} - \\
& 349.10130263 \times 538\ \text{nm} - 43.08258451 \times 576\ \text{nm} + 362.11769117 \times 577\ \text{nm} - \\
& 766.63930905 \times 578\ \text{nm} + 185.07826147 \times 579\ \text{nm} + 665.67198247 \times 580\ \text{nm} - \\
& 376.97383855 \times 581\ \text{nm} - 214.35928716 \times 596\ \text{nm} - 227.35866389 \times 597\ \text{nm} + \\
& 541.52766144 \times 598\ \text{nm} + 175.84690916 \times 608\ \text{nm} + 113.54131404 \times 609\ \text{nm} + \\
& 211.919296 \times 610\ \text{nm} - 594.98148968 \times 611\ \text{nm} - 51.3244556 \times 634\ \text{nm} + 284.70391609 \\
& \times 685\ \text{nm} - 306.84444538 \times 686\ \text{nm} + 1212.50847529 \times 701\ \text{nm} - 1936.03773524 \times 702 \\
& \text{nm} + 1314.09886101 \times 703\ \text{nm} - 559.33951346 \times 704\ \text{nm} + 41.3318793 \times 731\ \text{nm} + \\
& 85.77972942 \times 732\ \text{nm} - 6.39724394 \times 733\ \text{nm} + 2.57821684 \times 767\ \text{nm} - 1000.33835402 \\
& \times 768\ \text{nm} + 638.43201143 \times 769\ \text{nm} + 497.76415139 \times 784\ \text{nm} - 500.55432586 \times 785 \\
& \text{nm} + 178.4954138 \times 786\ \text{nm} + 278.09233576 \times 891\ \text{nm} - 97.70339466 \times 929\ \text{nm} + \\
& 17.59072331 \times 930\ \text{nm} - 139.22254262 \times 931\ \text{nm} - 338.49094149 \times 1100\ \text{nm} + \\
& 335.06610171 \times 1101\ \text{nm} + 192.94065592 \times 1109\ \text{nm} - 8.92408321 \times 1110\ \text{nm} - \\
& 215.20450248 \times 1111\ \text{nm} + 84.9204057 \times 1115\ \text{nm} + 105.04473744 \times 1149\ \text{nm} - \\
& 255.40995135 \times 1170\ \text{nm} + 510.86754127 \times 1248\ \text{nm} - 493.98138434 \times 1251\ \text{nm} + \\
& 131.95488055 \times 1255\ \text{nm} - 410.70269865 \times 1259\ \text{nm} - 425.9423493 \times 1264\ \text{nm} + \\
& 703.33488668 \times 1265\ \text{nm} - 880.17037685 \times 1273\ \text{nm} + 1217.18178423 \times 1274\ \text{nm} + \\
& 517.45423938 \times 1276\ \text{nm} - 2228.60786868 \times 1277\ \text{nm} + 1658.6689775 \times 1278\ \text{nm} + \\
& 405.04266763 \times 1280\ \text{nm} - 751.47374848 \times 1281\ \text{nm} + 1270.6998438 \times 1284\ \text{nm} - \\
& 1081.53285854 \times 1285\ \text{nm} - 94.52729087 \times 1288\ \text{nm} + 1303.6367605 \times 1293\ \text{nm} - \\
& 1595.58687516 \times 1294\ \text{nm} + 303.8508039 \times 1297\ \text{nm} + 219.35483385 \times 1300\ \text{nm} + \\
& 230.92135563 \times 1301\ \text{nm} - 857.25218971 \times 1304\ \text{nm} + 537.69713748 \times 1305\ \text{nm} + \\
& 190.37610831 \times 1308\ \text{nm} - 271.94718359 \times 1312\ \text{nm} + 543.88762491 \times 1418\ \text{nm} - \\
& 1825.22308523 \times 1421\ \text{nm} + 1399.02178773 \times 1422\ \text{nm} + 329.80032585 \times 1424\ \text{nm} - \\
& 4541.30142072 \times 1425\ \text{nm} + 4671.41597459 \times 1426\ \text{nm} - 578.23285744 \times 1434\ \text{nm} + \\
& 161.18160343 \times 1641\ \text{nm} - 116.28735521 \times 1645\ \text{nm} - 44.75511183 \times 1649\ \text{nm} - \\
& 15.22826404 \times 1889\ \text{nm} + 40.63028595 \times 2209\ \text{nm} - 72.16827033 \times 2218\ \text{nm} + \\
& 42.96064164 \times 2229\ \text{nm}
\end{aligned}
\tag{A4}
$$

$$
\begin{aligned}
\text{LogOlsenP Ravensdown Data} =\ & 1.41421696 - 262.38296498 \times 515\ \text{nm} + 258.3927565 \\
& \times 516\ \text{nm} + 172.63797279 \times 549\ \text{nm} + 565.81515751 \times 550\ \text{nm} - 760.83869552 \times 551 \\
& \text{nm} - 265.81557966 \times 611\ \text{nm} + 352.59398899 \times 612\ \text{nm} + 30.70039851 \times 613\ \text{nm} - \\
& 64.88346955 \times 614\ \text{nm} - 720.64113906 \times 626\ \text{nm} + 342.74442767 \times 627\ \text{nm} + \\
& 917.56280873 \times 628\ \text{nm} - 1258.58083917 \times 629\ \text{nm} + 838.638906 \times 630\ \text{nm} - \\
& 977.90498588 \times 662\ \text{nm} + 1791.20280789 \times 663\ \text{nm} - 1142.04728078 \times 664\ \text{nm} + \\
& 446.0679803 \times 689\ \text{nm} - 148.19112984 \times 690\ \text{nm} - 232.03507741 \times 708\ \text{nm} - \\
& 11.00376379 \times 709\ \text{nm} + 252.55706692 \times 718\ \text{nm} - 4167.75593136 \times 727\ \text{nm} + \\
& 7798.83249416 \times 728\ \text{nm} - 6592.26457503 \times 729\ \text{nm} + 2897.4217993 \times 730\ \text{nm} + \\
& 240.60451648 \times 757\ \text{nm} - 107.42720557 \times 758\ \text{nm} + 258.77876235 \times 759\ \text{nm} - \\
& 471.60005204 \times 762\ \text{nm} - 748.4786656 \times 802\ \text{nm} + 951.61763404 \times 803\ \text{nm} - \\
& 167.32254136 \times 804\ \text{nm} - 797.01853402 \times 866\ \text{nm} + 857.16628768 \times 867\ \text{nm} - \\
& 76.77805452 \times 868\ \text{nm} + 217.55905357 \times 1178\ \text{nm} - 246.85313641 \times 1179\ \text{nm} + \\
& 22.81698457 \times 1232\ \text{nm} + 105.3998638 \times 1483\ \text{nm} + 1005.05780092 \times 1484\ \text{nm} - \\
& 621.61522117 \times 1485\ \text{nm} - 4286.12068281 \times 1486\ \text{nm} + 4315.85298814 \times 1487\ \text{nm} + \\
& 1035.3165668 \times 1488\ \text{nm} - 2177.24935658 \times 1489\ \text{nm} + 1891.09900621 \times 1490\ \text{nm} - \\
& 1646.62280566 \times 1491\ \text{nm} + 2219.08246915 \times 1492\ \text{nm} - 2146.86066432 \times 1493\ \text{nm} + \\
& 319.98869202 \times 1494\ \text{nm} - 11.55010056 \times 1793\ \text{nm} - 187.35487266 \times 1886\ \text{nm} + \\
& 411.3407968 \times 1887\ \text{nm} - 155.68803623 \times 1888\ \text{nm} - 230.23052205 \times 1898\ \text{nm} + \\
& 154.42632413 \times 1899\ \text{nm} - 49.4719615 \times 1907\ \text{nm} + 258.06274043 \times 1908\ \text{nm} - \\
& 1130.31963003 \times 1911\ \text{nm} + 1195.36384891 \times 1912\ \text{nm} - 285.57576757 \times 1914\ \text{nm} + \\
& 33.27687837 \times 1916\ \text{nm} + 35.10204655 \times 1925\ \text{nm} + 6.68288051 \times 1940\ \text{nm} - \\
& 146.78208608 \times 1941\ \text{nm} + 66.67246198 \times 1945\ \text{nm} + 40.53099809 \times 1956\ \text{nm} + \\
& 3.85709384 \times 1989\ \text{nm} - 1.53557831 \times 1993\ \text{nm} - 750.56274736 \times 2068\ \text{nm} + \\
& 906.91294082 \times 2069\ \text{nm} - 453.9769159 \times 2070\ \text{nm} + 238.11151817 \times 2071\ \text{nm} + \\
& 355.80990021 \times 2083\ \text{nm} - 314.89494526 \times 2084\ \text{nm} + 32.22063511 \times 2192\ \text{nm} - \\
& 41.55074839 \times 2193\ \text{nm} - 63.68003103 \times 2360\ \text{nm} + 248.87248573 \times 2364\ \text{nm} - \\
& 191.41192263 \times 2365\ \text{nm} + 37.99319427 \times 2368\ \text{nm} + 6.5000641 \times 2382\ \text{nm} - \\
& 97.59834886 \times 2385\ \text{nm} + 122.06277083 \times 2386\ \text{nm} - 149.60606473 \times 2389\ \text{nm} + \\
& 99.81677335 \times 2390\ \text{nm} - 18.9263297 \times 2394\ \text{nm} - 75.95234572 \times 2426\ \text{nm} + \\
& 413.74670128 \times 2427\ \text{nm} - 305.23617967 \times 2428\ \text{nm} - 170.12695893 \times 2429\ \text{nm} + \\
& 224.38494028 \times 2430\ \text{nm} - 345.81770354 \times 2431\ \text{nm} + 328.5991858 \times 2432\ \text{nm} + \\
& 144.86200043 \times 2433\ \text{nm} - 197.93334943 \times 2434\ \text{nm} - 9.76486789 \times 2450\ \text{nm} + \\
& 28.00901871 \times 2469\ \text{nm} - 15.8530771 \times 2470\ \text{nm}
\end{aligned}
\tag{A5}
$$

$$
\begin{aligned}
\text{Log Olsen P Massey Data} = {}& 0.88013594 - 17.65266195 \times 515\ nm + 32.64831668 \times 516 \\
& nm - 23.87045858 \times 549\ nm - 24.62521319 \times 550\ nm + 11.81183849 \times 551\ nm - \\
& 16.76072924 \times 611\ nm - 19.90858466 \times 612\ nm + 39.71002999 \times 613\ nm + 24.8627212 \\
& \times 614\ nm + 56.8155205 \times 626\ nm - 13.74100331 \times 627\ nm - 29.22353436 \times 628\ nm + \\
& 5.16801954 \times 629\ nm + 21.54317538 \times 630\ nm + 2.35543078 \times 662\ nm - 134.35829477 \\
& \times 663\ nm + 4.10338132 \times 664\ nm + 93.51692756 \times 689\ nm + 10.70528884 \times 690\ nm + \\
& 5.94483344 \times 708\ nm - 17.02253754 \times 709\ nm - 37.71109024 \times 718\ nm + 30.38784528 \\
& \times 727\ nm - 6.34950395 \times 728\ nm + 12.8813182 \times 729\ nm - 20.96765759 \times 730\ nm + \\
& 53.39606312 \times 757\ nm - 11.07370831 \times 758\ nm + 29.08119915 \times 759\ nm - \\
& 18.69580045 \times 762\ nm - 43.03481018 \times 802\ nm + 10.56616882 \times 803\ nm - \\
& 49.96158803 \times 804\ nm - 11.93210691 \times 866\ nm + 69.1725134 \times 867\ nm - 8.91672726 \times \\
& 868\ nm - 24.96420098 \times 1178\ nm - 17.85061366 \times 1179\ nm + 38.48982714 \times 1232\ nm \\
& + 34.2186828 \times 1483\ nm + 1.71967816 \times 1484\ nm + 35.11784548 \times 1485\ nm - \\
& 45.39746462 \times 1486\ nm - 3.33519592 \times 1487\ nm - 21.37228564 \times 1488\ nm + \\
& 5.09598897 \times 1489\ nm - 5.29940748 \times 1490\ nm + 2.53762136 \times 1491\ nm + 3.17278458 \\
& \times 1492\ nm + 9.30919722 \times 1493\ nm - 31.51731844 \times 1494\ nm - 2.73099208 \times 1793\ nm \\
& - 7.66321304 \times 1886\ nm + 31.02393364 \times 1887\ nm - 0.48538538 \times 1888\ nm - \\
& 69.04625392 \times 1898\ nm + 9.87413403 \times 1899\ nm + 35.69223335 \times 1907\ nm + \\
& 15.37528336 \times 1908\ nm + 46.16127723 \times 1911\ nm + 24.82627982 \times 1912\ nm - \\
& 9.04673896 \times 1914\ nm - 57.03018431 \times 1916\ nm + 8.94977846 \times 1925\ nm - \\
& 61.10786769 \times 1940\ nm - 1.22981847 \times 1941\ nm + 59.27526757 \times 1945\ nm - \\
& 13.94808084 \times 1956\ nm - 23.16149111 \times 1989\ nm + 24.13821412 \times 1993\ nm - \\
& 39.09155072 \times 2068\ nm + 7.04608705 \times 2069\ nm + 7.43490488 \times 2070\ nm + \\
& 55.21290205 \times 2071\ nm - 14.29909075 \times 2083\ nm - 4.95445671 \times 2084\ nm - \\
& 31.51479668 \times 2192\ nm + 39.08059 \times 2193\ nm + 11.90398454 \times 2360\ nm - \\
& 35.53245923 \times 2364\ nm - 33.02879699 \times 2365\ nm + 61.63470833 \times 2368\ nm + \\
& 10.47308603 \times 2382\ nm + 9.11044008 \times 2385\ nm + 12.24158109 \times 2386\ nm - \\
& 20.01093487 \times 2389\ nm + 42.05763612 \times 2390\ nm - 56.47962304 \times 2394\ nm - \\
& 56.49844962 \times 2426\ nm + 47.71415088 \times 2427\ nm - 17.90993997 \times 2428\ nm + \\
& 43.29705927 \times 2429\ nm + 33.88853302 \times 2430\ nm - 80.79551169 \times 2431\ nm + \\
& 39.38333687 \times 2432\ nm + 44.5568069 \times 2433\ nm - 104.62940795 \times 2434\ nm + \\
& 61.64147234 \times 2450\ nm - 8.92227817 \times 2469\ nm - 22.35364028 \times 2470\ nm
\end{aligned}
\tag{A6}
$$

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
