# Peer review of "Technical Note: Regression Analysis of Proximal Hyperspectral Data to Predict Soil pH and Olsen P"

_agriculture, doi:10.3390/agriculture9030055_

Reviewer 1 Report

Overall comments

The scope of this manuscript is relevant and would be of interest to readers of Agriculture. The technique the authors described in this manuscript has a good intention to utilize the hyperspectral data to predict soil pH and available phosphorus. However, I have concerns about this manuscript. First of all, the abstract needs major improvement; not only regarding the order of sentences but also in vague wording. Second, the ‘introduction’ is not convincing to justify the importance of this work. The second paragraph of the introduction lightly addresses the significance of the manuscript. However, it is not enough information for the international audience. The information in the paragraph would be enough for the people who live near Massey University or hill country farms but is not enough for people who are not familiar with Massey University or hill country farms. Third, in ‘Materials and Methods’, the authors indicated this study uses a total of 3190 soil samples from private and government research program. I wonder the authors took a consensus from the program or not, which is not clear from the manuscript. Forth, I do not think authors need to describe the detailed protocol of Olsen P because it is a part of standard method unless the authors modified a part of the protocol. Fifth, the description of results seems too short to understand for the wider audience. Sixth, the logical flow in the discussion is abrupt and difficult to follow. For example, it was not clear which wavebands are effective for the estimation. Larger factor numbers (in this case wavebands) usually results in high R2 number and the larger number of samples easily provide lower p-values.

In details

L21: ‘very good prediction’ is too vague to be explicit. Please revise.

L22: ‘very significant correlation but with much more noise’ could be misleading. Especially this result ‘may indicate’ the conclusion.

L23-24: This sounds a very unprofessional excuse. Suggest eliminating.

L28-29: This sentence should be placed toward at the end of the introduction.

L29-31: The information here is in the affiliation. Redundant information for the audience.

L33: ‘hill country farms’ is too vague for the international audience. This could be everywhere.

L36: Transition of the sentences is too abrupt. It is difficult to follow.

 L40-48: This paragraph does not have a smooth logical flow, which provides a not convincing justification of the work.

L45: ‘accurately measured’ … what was the accuracy of the measurements?

L51: It seems there are 2 spaces after ‘Olsen P.’

L58: Locations of ‘three farms’ are not clear.

L58-59: I assume there is no issue on the IP of the data. It should be clearly stated.

L61-62: ‘The easiest soil properties to be’ should be eliminated.

L63-64: It is not clear which farm of the three farms were used.

L77: ‘In preparation … strongly acidic reagent’ is unnecessary information.

L83-84: Need to show the ratio of soil:water, because the ratio is different by the country (1:1, 1:2, 1:5, 1:10 etc).   

L87-89: The information here is redundant from the paragraph. Suggest eliminating.

L94-95: ‘requires powerful statistical software’ is unnecessary information.

L96-97: This information is an unnecessary excuse for the international audience.

L99: ‘(Signif. …. ‘*’ 0.05)’ is unnecessary information.

L111-112: Was this log10 transformation for passing the normality test?

L148: ‘good results’ is vague; this should be more explicit.

L152: ‘reasonable result’ is vague; this should be more explicit.

L152-156: Coexisting method, result, and discussion does not clearly show the point of argument.

L157: ‘a better prediction’ is vague; this should be more explicit.

L162-164: This point of the argument is misleading for the international audience.

L168: From the provided data, this argument is questionable.

L169: This statement is misleading without cost data.

Author Response

All suggestions by reviewer 1 have been accepted and revised.

L21 Revised

L22 revised

L 23-24 Revised eliminated as suggested

L28 - 29 revised

L29 - 31 revised but in conflict with reviewers 2 and 3 who wanted the lab named.

L 33 A description of the farming system and map of the farms has been included and the farm names and locations has been provided and permitted to publish.

L 36 altered as suggested

L 40 -46 revised

L45 revised

L 51 revised

L58 addressed in L33

L58 -59 addressed IP released

L 61 - 62 eliminated as requested

L 63 - 64 addressed already L33 and L58

L77 the method has been eliminated as same as in [7] at suggestion of editor and Reviewer 3.

L83 :84 water dilution rate added 10g of soil to 25ml of distilled water

L87 - 89 eliminated

L94 - 95 eliminated as requested

L96 -97 eliminated as requested

L99 eliminated as requested

L111 - 112 yes information and shape of data sets added as both Reviewer 1 and 3 made this comment.

L148 actual figures now reported as requested

L 152 actual figures now reported

Lines 152 - 188 Discussion rewritten to address points raised by reviewers 1 and 3.

Reviewer 2 Report

Review of the manuscript  Regression Analysis of Proximal Hyperspectral Data to Predict Soil pH and Olsen P

Remarks

It seems to me that the analyses described in  manuscript give a potential for  quite ambitious work. I can even find sometimes some positive points in the manuscript. However, the manuscript is unclear, and so poorly written, that I should state, with great regret, that the manuscript at present form does not meet the standards of good scientific journal as Agriculture. 

Therefore, I should state that the manuscript cannot be published in Agriculture in present form, and should be greatly revised or resubmit.

However, I encourage the Authors to perform the major revision/ resubmision. Below, I will enumerate shortly the greatest shortcomings of the manuscript. There is no use to explain the issue that should be clear for scientists.

Major remarks

1.   The editorial quality (graphs, equations, captions quality of the text – just one example “The regression analyses undertakeni” etc.) are out of common standards.

2.   The experimental data sets are not described and presented sufficiently (summary statistics of soil  parameters etc.). This should be very important part of the work (I would suggest separate section). All soil parameters should be clearly described, and the deeper discussion on the influence of main soil parameters should be made

3.   The calculations based on PLSR should be supplemented by other statistical methods, for example other types of regression. In fact the statistical analysis is to scarce, there is no description of outliers

4.   The analysis of the impact of  the size of the data set on regression results should be carefully studied.

5.   The references about spatial variability of soil contamination/soil parameters/ fertility indicators  [2-4] are rather too basic. There are newer and deeper examples how to take into account this variability using geophysical measurements (e.g. Magnetometric assessment of soil contamination in problematic area using empirical Bayesian and indicator kriging: A case study in Upper Silesia, Poland, P. Fabijańczyk, J. Zawadzki, T. Magiera, Geoderma 308, 69-77).

6.   The process of removing/selecting unnecessary bands should be described in detail, the bands which were left (“the significant spectra found for a partial least square regression (PLSR)”should be described better. This process should be discussed in detail, because it is also essential result of the study.

7.   The experiment should be presented very more detaily, especially the spectra measurements. The experimental problems which might influence results should be underlined.

8.   The discussion section has the literature references, the same as in introduction, and is filled often with the information which can be found earlier.

9.   R2, Adj. R2 are very rough parameter of fitting quality. Other methods should be employed as for example confidence and prediction bands.

10.               And last but not the least: in conclusion Authors write “The exercise shows…”. However, this should be a scientific manuscript, not just an exercise. Hence, revise the manuscript accordingly, please. You have a lot of good experimental data.

Sincerely yours

Reviewer

Author Response

These have been significantly revised, charts farm descriptions added. The 16,000ha of farms have hundreds of soil types which makes describing the soil types not possible.

The calculations, shape of data sets and descriptions updated. the budget for this work is exhausted further extensive analysis will need to be undertaken by others, equations describing wavelengths found have been included.

see 2

Discussion around this has been added

Thanks for the information, again 16,000 hectares of farms much of which is impassable by vehicle and with grazing slopes up to 40 degrees makes use of this equipment impractical. The soil samples are taken as per methods [2 -4] the larger data set as per [2] the smaller data set as per [4].

"R" rates each of the 2,150 bands by signicance the most significant 100 bands were selected based on this information.

More detail has been added.

The discussion has been rewritten to reflect comments from reviewers 1,2 and 3.

More information has been added this was a search for bands which may indicate Olsen P and pH on a limited budget, more information is supplied.

This was a specific exercise based on a limited budget which is exhausted. However, it is hoped that now the equations, wavebands, and findings are fully released others may carry on the work.

Reviewer 3 Report

The paper entitled “Regression analysis of proximal hyperspectral data to predict soil pH and Olsen P” submitted to Agriculture used regression analysis on a large data set (n =3190 soil samples) to determine the utility of predicting soil pH and Olsen extractable P from near infrared (NIR) data. Such research may be important, since NIR-based estimates of pH and Olsen P could be more cost effective than traditional wet chemistry methods, provided NIR estimates are sufficiently accurate and precise. In general, the writing makes it difficult to understand the article’s main points. I suggest a thorough rewriting to better clarify the utility and novelty of the research and how the findings could be applied to improve cost-effectiveness of soil pH and Olsen extractable P as traditionally measured. I have included comments and suggestions directly on the manuscript in an effort to help the authors frame their revision.

Author Response

All the suggestions and comments made by reviewer 3 have been implemented.

The only one that it was not possible to address was moving the axes of actual and fitted data. the graphs are produced by StatTools which graphs the independent variable fitted data to the y axes.

The equations are in Appendix 1.

Round  2

Reviewer 1 Report

The revised manuscript is improved quite a lot compared to the previous version. However, there are still issues in this manuscript. Please see below for my comments.

Abstract:

L45: ‘see Figure 1’should be eliminated because the figure does not show what the sentence described.

L60, L62: Suggest changing ‘may prove to be useful’ to ‘has potential.’

L61-63: This sentence is out of place. Identifying spectra bands is a very important part of this article; however, the results do not show the reason for band selection (I assume the selection was done with the stepwise procedure). Also, the discussion does not address this issue.

L71-72: This sentence is out of place.

L74-77: Suggest combining with the previous paragraph. The sentence in L75-77 could move before the sentence L74-75.

L101-103: The sentence does not flow well. Please revise.

L106: Suggest moving this sentence to L121-122, or move up L121-122 to L106.

L122: Does ‘improve normality’ mean the dataset passed the normality test?

L123-128: The reason for the selection of waveband is weak. As is authors choose the bands based on the limitation of the software (StatTools). The selection of variables should be based on stepwise procedures.

L134-135: Rather than this pointing sentence, suggest to add the brief description of the outcome of the regression analysis.

L135: This sentence is overkill. The use of the software is already described in the method section.

L142-143: Pointing sentence. Please revise.

L143-L149: Rather than the general comments of the five figures, it is better to have a more specific description of results per figure.

L151-L159: For Y-axis of all figures, suggest to use ‘model estimates’ than ‘Fit.’ Also, the figures are easier to understand if the shape was square with the same value range in both X&Y axes rather than the rectangle with a different range of values between X and Y. Also, adding a 1:1 line in all figures would help to understand the accuracy of the model.

L156: [Figure 2-c] Please make one of the lines to dash to make it easier for the audience to understand. Also, suggest performing slope analysis to check if the two slopes are significantly different or not.

L159: [Figure 2-d] same comment as the previous one.

L172-173: Statement in the ( ) is out of place. Please revise. Maybe better to put in the second sentence of the paragraph (L174-175).

L176-180: This kind of justification for the work should be in ‘Introduction’ rather than here.

L181-182: Please revise the sentence to clarify the point.

L185-188: It is not clear how this sentence relates to the topic sentence of the paragraph.

L188-190: I agree with ‘the P is immobile in soil’; however, it is hard to believe that the phosphate is immobile mainly because of the cation exchange. Cation exchange is an indirect factor for P-fixation. Fe and Al oxides in clay minerals and organic matter are direct factors for the P-fixation.

L195-198: Normal distribution of the dataset is a presumption of the regression analysis. This sentence would be better in the ‘Results’ section.

L199-201: This paragraph belongs to the ‘Results’ section.

L202-205: These sentences belong to the ‘Methods’ section.

L205-206: The slope analysis between the two gradients would indicate better reason than the observational reason.

L206-207: If the slope analysis results in a significant difference between the two gradients, then the spectral wavebands are not transferable. Also, validation of the model for one location only is too weak to make this statement.

L210-211: Showing the cost of hyperspectral sensing would help the audience to understand how much cheaper than the soil tests.

In the abstract, authors mentioned 'the 100 most significant wavebands'. I feel this point should be discussed with the plausible reasons and possible underlying mechanism for the relationships of hyperspectral wavebands and Olsen P or pH.

Author Response

A new figure 1 has been added.

Revised sentence

Selection of wave bands was by most significant and comment entered.

Altered figures did not add a 1-1 line as too close to one of the other lines and was confusing. Added analysis with log10 of both data sets and undertook normality test.

dash lines added

changed all sentences and made all revisions as requested.

Plausible reasons added.

Reviewer 2 Report

Dear Authors,

now the work is better, although still not very good. Especially, regarding geostatistics consider the remark I made in point 5 of my previous review, but I leave it to your decision.

Best regards,

Reviewer

Author Response

Work revisd, normality test added.

Reviewer 3 Report

Please refer to edits made directly on the PDF copy of the manuscript. 

Author Response

Commas added and deleted as requested.

Normality test added

wording revised as requested.

Introduction with farming type as requested by other reviewers

equations requested as by another reviewer.